# Seed Shape Description and Quantification by Comparison with Geometric Models

**Emilio Cervantes *** and **José Javier Martín Gómez**

IRNASA-CSIC, Cordel de Merinas, 40, E-37008 Salamanca, Spain
* Correspondence: emilio.cervantes@irnasa.csic.es; Tel.: +34-9-2321-9606

**Abstract:** Modern methods of image analysis are based on the coordinates of the points making the silhouette of an image and allow the comparison between seed shape in different species and varieties. Nevertheless, these methods miss an important reference point because they do not take into consideration the similarity of seeds with geometrical figures. We propose a method based on the comparison of the bi-dimensional images of seeds with geometric figures. First, we describe six geometric figures that may be used as models for shape description and quantification and later on, we give an overview with examples of some of the types of seed morphology in angiosperms including families of horticultural plants and addressing the question of how is the distribution of seed shape in these families. The relationship between seed shape and other characteristics of plant species is discussed.

**Keywords:** geometric curves; J index image analysis; morphology; seed; shape

## 1. Introduction

Shape is an important property of plants that has been used for the description of organs and structures since the origins of botany. Some plant species were termed on the basis of the shape of their leaves (for example *Drosera rotundifolia*, *Plantago ovata*), others according to the shape of their fruits (*Coronilla scorpioides*, *Eugenia pyriformis*), and others following the forms of their seeds (*Vicia platysperma*). Seed shape is an interesting property that has been used in the taxonomy of the Caryophyllaceae [1–3], Orchidaceae [4], Papilionaceae [5], and Zingiberaceae [6].

Seed shape is related to dispersal. The potential for dispersal is increased by the growth of extra-ovular structures such as plumes, wings, hooks, oil bodies, and diverse types of arils [7] that complicate the analysis of seed morphology. But, leaving out such structures, seed description may be based on similarity to geometrical figures such as the oval, ellipse, cardioid, and others. The comparison of seed images with these figures gives a precise description that can be quantitative, allowing the comparison between genotypes or seeds grown in different conditions.

Recent methods for the comparison of seed shapes depend on high-throughput systems based on digital image analysis [8,9]. The methods combine artificial vision technologies with statistical algorithms and may be very efficient for the discrimination of seeds and fruits in a range of plant species and varieties [10–13]. Nevertheless, these methods have two main drawbacks: (1) Their results do not give a magnitude that can be associated with seed shape for a particular group of seeds, and (2) they do not take into consideration the similarity of seeds with geometrical figures, missing an important reference point. This has resulted in a "gap" in the current scientific literature on seed shape. Current methods based on artificial vision, algorithms, and statistical analysis need to go back to the initial seed images, combining geometric models with quantitative and statistical methods to analyze seed shapes.

In this review, we first describe the geometric models that may be applied for seed shape quantification in diverse plant families including important horticultural plants. Then, we present some examples of families where the models have been applied and discuss possible models in families of interest in horticulture. Finally, we discuss the relationship between seed shape and other plant characteristics and the potential of seed morphology in taxonomy.

## 2. Geometric Models

Figure 1 presents a summary of the geometric models used for the description and quantification of seed shape. These are: The cardioid and derivatives; the ellipse; the oval; the contour of Fibonacci's spiral, heart-shaped curves, and lenses of varied proportions.

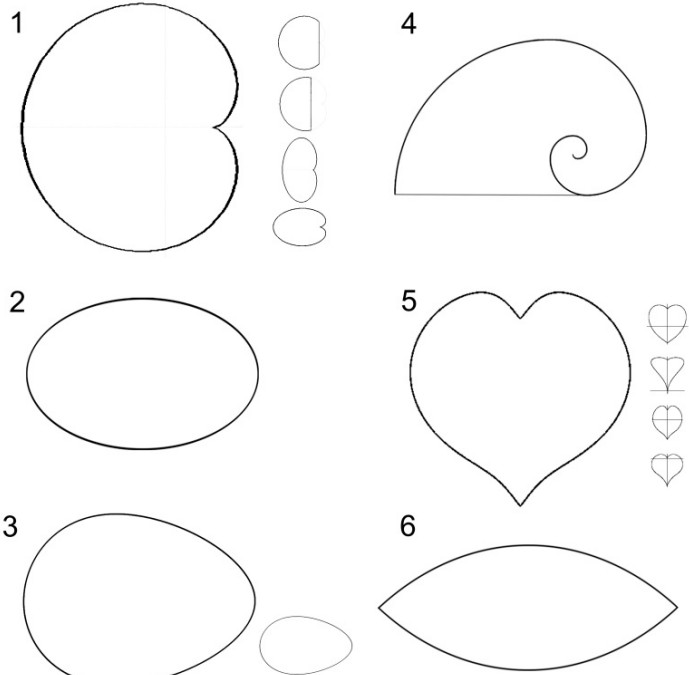

**Figure 1.** Main models used in the description and quantification of seeds. (**1**) Cardioid and derivatives; (**2**) ellipse; (**3**) oval and elongated oval; (**4**) the contour of Fibonacci's spiral; (**5**) heart-shaped curves; (**6**) lens.

In this section, we describe each of these models. The origin of the images used is indicated in the Appendix A.

### 2.1. Cardioid

The cardioid is the curve traced by a point on the perimeter of a circle that is rolling around a fixed circle of the same radius. Another definition is that of the envelope of a pencil of circles formed in a way that their mid points lie on the perimeter of the fixed generator circle [14]. The cardioid has been long recognized as a tool to describe biological structures during development because it corresponds to the figure generated by an organ that grows from a fixed point of insertion [15]. Thus, plane figures of animal embryos, leaves, and seeds resemble the cardioid (Figure 2).

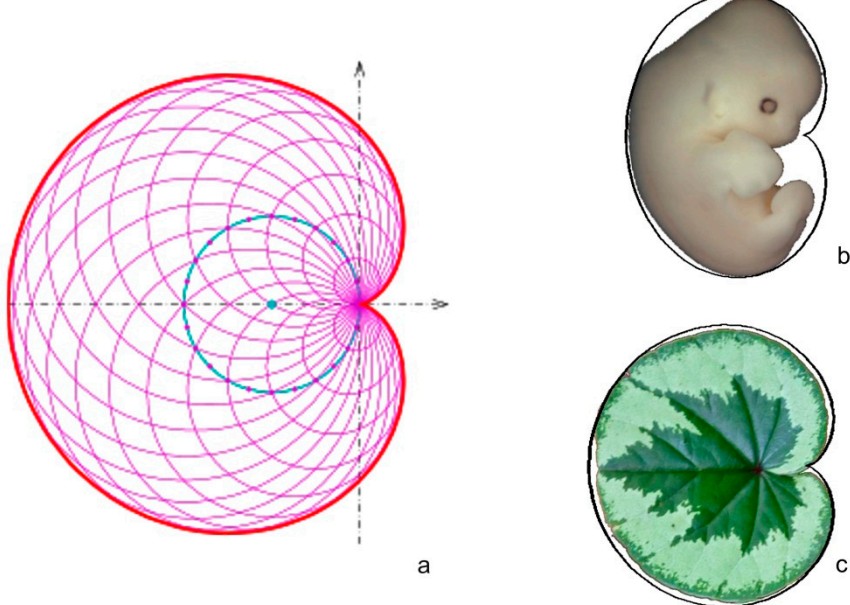

**Figure 2.** (**a**) The cardioid visualized as the envelope of a pencil of circles formed in a way that their midpoints lie on the perimeter of the fixed generator circle. The cardioid shape is common in embryos (**b**) and leaves (**c**).

## 2.2. Ellipse

The ellipse is defined as a regular oval shape, traced by a point moving in a plane so that the sum of its distances from two fixed points (the foci) is constant, or resulting when a cone is cut by an oblique plane that does not intersect the base [16]. An ellipse is defined by the length of the major and minor axes. Changing the relations between the axes results in different ellipses, more or less elongated. Leaves of many species tend to an elliptical shape, which is also common among diverse animals like for example *Stylostomum ellipse*, a platelmynth (Figure 3) [17].

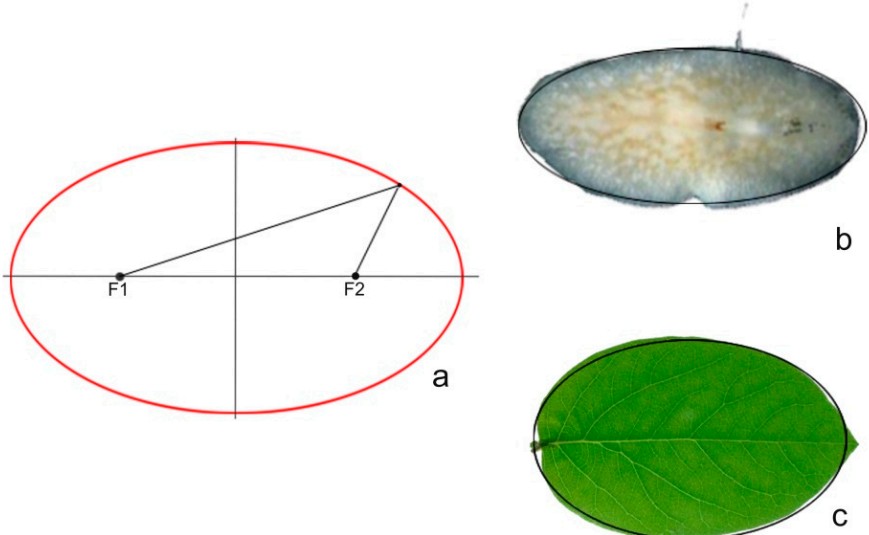

**Figure 3.** (**a**) The ellipse is a regular oval curve, traced by a point moving in a plane so that the sum of its distances from two other points (the foci, F1 and F2) is constant. The ellipse is observed in the image of a platelmynth (**b**) and in leaves of many species (**c**).

### 2.3. Oval

An oval is a curve resembling a squashed circle but, unlike the ellipse, without a precise mathematical definition. The word oval is derived from the Latin word "ovus" meaning egg. Unlike ellipses, ovals have only a single axis of reflection symmetry (instead of two). The oval may be obtained by the combination of four arcs of circle, one of which is a semi-circle (AB), the other two have their centers in the extreme points of the semicircle (A, B), and the fourth arc closes the image with a quarter of circle traced from the point (C), equidistant to A and B [18]. Typical ovals are observed in eggs and in some fruits such as avocado (Figure 4).

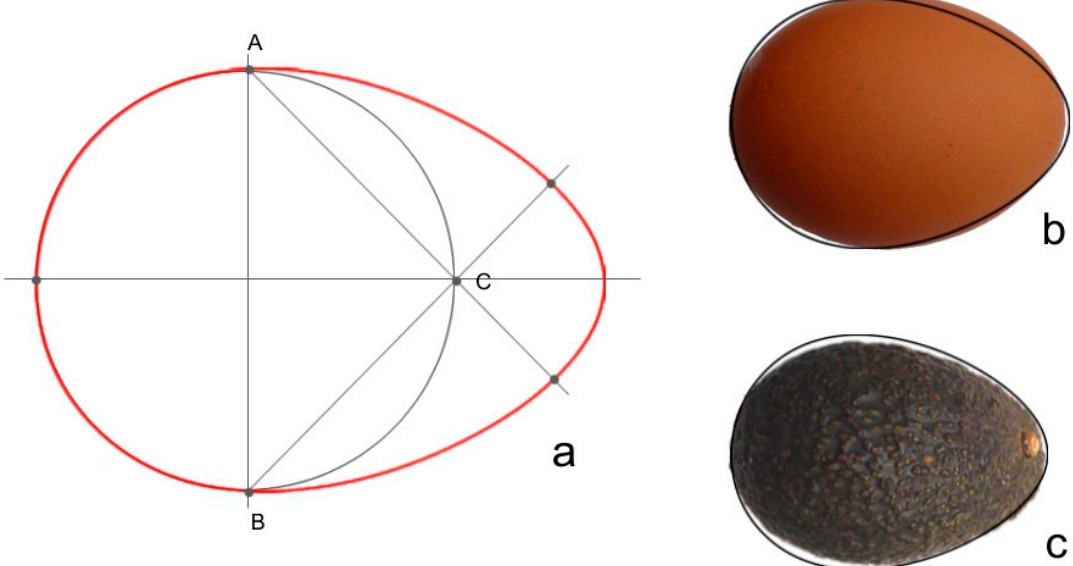

**Figure 4.** (**a**) The oval is a curve resembling a squashed circle but, unlike the ellipse, without a precise mathematical definition. In the image, the oval is obtained as a combination of four arcs of circle. The oval is observed in eggs (**b**) and avocado fruit (**c**).

### 2.4. Contour of Fibonacci's Spiral

Successive points dividing a golden rectangle into squares lie on a logarithmic spiral, which is sometimes known as the golden spiral [19,20] (Figure 5). The logarithmic spiral is a spiral whose polar equation is given by

$$r = ae^{(b\,\theta)}$$

where r is the distance from the origin, θ is the angle from the x-axis, and a and b are arbitrary constants [21].

### 2.5. Heart Curve

There are a number of mathematical curves that produced heart shapes, some of which are illustrated below (Figure 6) [22].

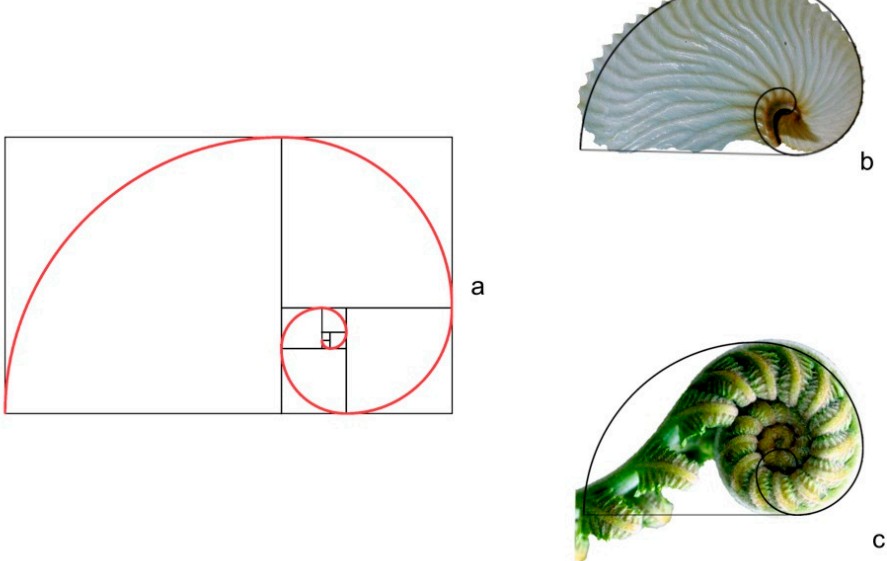

**Figure 5.** (**a**) Construction of a Fibonacci spiral. The Fibonacci spiral is observed in sea shells of the species *Argonauta argo* (**b**) as well as in the growth of some species of ferns (**c**).

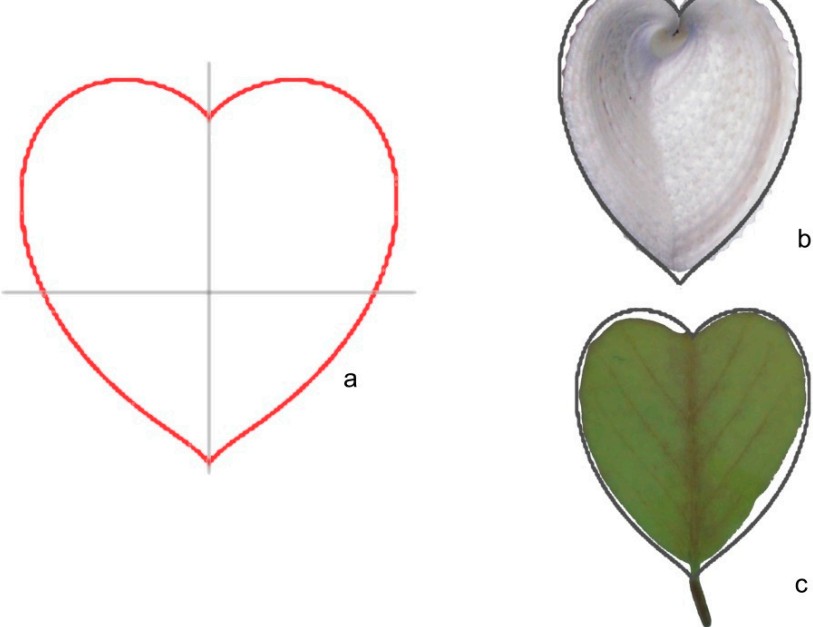

**Figure 6.** (**a**) An example of heart-shaped curve (**a**) and similar images in sea snails of the species *Corculum cardissa* (**b**) and leaves of alfalfa (*Medicago sativa*) (**c**).

*2.6. Lens*

A lens is a lamina formed by the intersection of two offset disks of unequal radii such that the intersection is not empty, one disk does not completely enclose the other, and the centers of curvatures are on opposite sides of the lens. Images of some fishes as well as fruits resemble lenses (Figure 7). If the centers of curvature are on the same side, the resulting figure is called a lune [23].

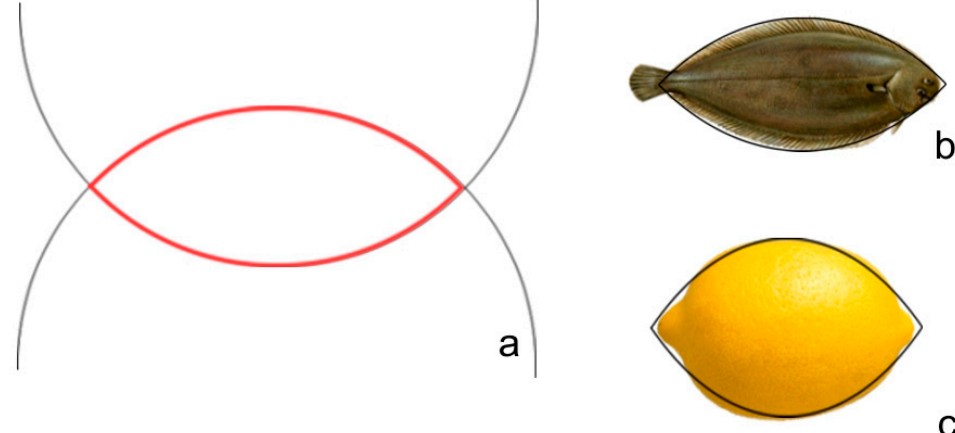

**Figure 7.** (**a**) The lens is the figure produced by the intersection of two circles. (**b**) *Solea solea* sp., a fish of the family Soleidaidae. (**c**) A lemon fruit (*Citrus* sp. Rutaceae).

## 3. Models Predominant in Particular Species and Taxonomic Groups

### 3.1. Variations in Seed Shape

In general, seed shape is conserved at the species level. Nevertheless, shape is the result of a growth process and even in the same plant, or also in the same fruit, there may be differences in the shape of seeds depending on many factors such as the ongoing developmental status or the position of the seed in the plant, in the inflorescence or in the fruit, and depending as well on the type of fruit. In fruits forming aggregates of seeds, such as in some capsules, the shape of each individual seed depends upon its position. This occurs in species of the Meliaceae (for example *Swietenia mahogany* Jacq.), Myrtaceae (*Eucalyptus* L'Hér.) [24], and Nitrariaceae (*Peganum harmala* L.) (Figure 8), as well as in Nictaginaceae and other families.

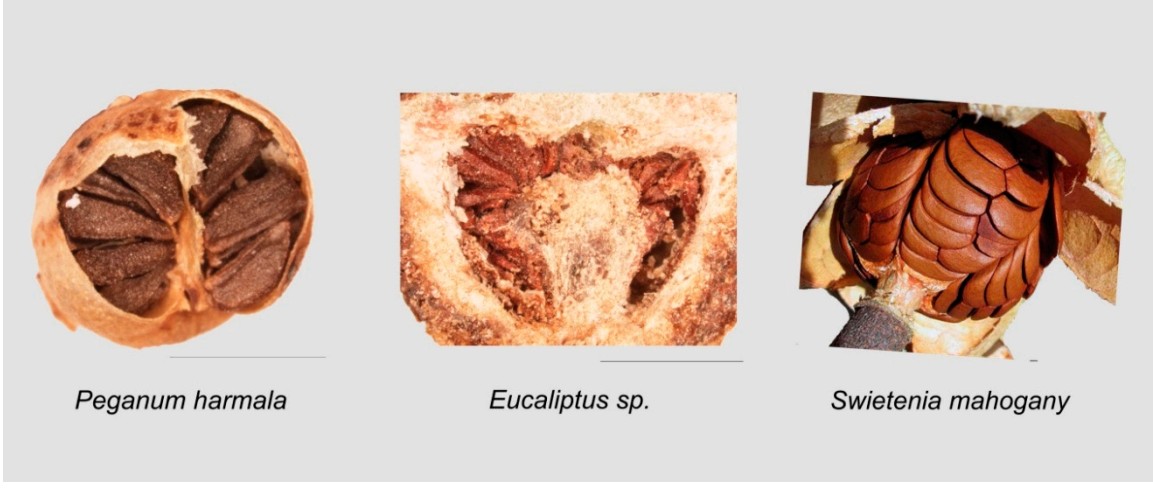

**Figure 8.** In fruits forming aggregates of seeds, the position of a seed in the aggregate determines their shape. Examples: *Peganum harmala* L., (Nitrariaceae); *Eucalyptus* sp. L'Hér. (Myrtaceae); *Swietenia mahogany* Jacq. (Meliaceae).

Intra-specific variability in seed shape may be due to causes other than differences between seeds of the same plant. Differences between seeds may occur between wild-type and cultivated subspecies and varieties, as reported for *Triticum aestivum* L. [25]. In wheat, differences in shape have been reported between ancestral genotypes (ssp. *spelta* (L.) Thell., sive *Triticum spelta* L.) and modern bread subspecies and varieties (ssp. *aestivum*, cv. Torke or Zebra for example), the latter being more rounded [25,26].

Also, seed shape may vary in plants grown in different geographic regions or climatic conditions, as observed in *Olea europaea* L. [27], *Jatropha curcas* L. [28], and *Ricinus communis* L. [29,30]. Finally, quantitative differences in seed shape between wild-type seeds and mutants have been also reported for *Arabidopsis thaliana*, *Medicago truncatula*, and *Lotus japonicus* [31–33]. Nevertheless, and taking into account all these cases of variation, seed shape may be considered relatively constant for the majority of species under normal growth conditions. In contrast, there may be large differences in seed shape between species of the same family, but, also in some families, a model may be predominant. Even in some orders, most of the genera and species may have their seeds resembling a few particular models.

In the following sections, we review the cases where particular geometric figures have been used for the description and quantification of seed shape in some species and higher taxonomic groups, allowing the analysis of seed shape with practical applications. Cardioid-derived figures were used in the model plant *Arabidopsis thaliana* (L.), Heynh. (Brassicaceae) [32], as well as in the model legume *Medicago truncatula* Gaertn. (Fabaceae) [33,34]. The cardioid was also the figure used in the model legume *Lotus japonicus* L. (Fabaceae) [33], and in *Capparis spinosa* L. (Capparaceae) [35]. The ellipse was the model in *Jatropha curcas* L. and *Ricinus communis* L. (Euphorbiaceae) [28–30], and in *Quercus* species (Fagaceae) [36].

We also review the cases of some families that have seeds resembling a predominant model or in which the application of a model has allowed a new point of view on taxonomic relationships. Finally, we discuss the cases where a given morphotype is abundant in an order. The oval and the ellipse are the predominant types in the Cucurbitales and the heart-shaped curve the predominant type in the Vitales.

### 3.2. The Cardioid as a Tool for Seed Shape Quantification in Model Species

The magnitude used to describe the similarity between the geometric model (cardioid) and the seed image is called J index. J index is the percent of similarity between both images, geometric model (cardioid), and seed image. It is the percent of the total surface of both images that is shared (ratio shared/not shared ×100; see, for example, [32–35,37]). *Arabidopsis* seeds resemble cardioids elongated in the direction of the symmetry axis by a factor of Phi (the Golden Proportion). Their values of J index are over 90 and increase in the course of imbibition before germination, reaching maximum values of 95 and over at 20 h imbibition [38]. Differences in seed shape revealed as differences in J index were described with reduced values respect to the wild type for mutants in both the ethylene signal transduction [32] and cellulose biosynthesis [38] pathways in *Arabidopsis thaliana*. Similarly, seeds of the model legumes *Lotus japonicus* and *Medicago truncatula* were found to resemble, respectively, the cardioid and the cardioid elongated by a factor of Phi. Values of J index increase in the course of imbibition [34] and differences were found between wild-type seeds and ethylene pathway mutants in both species [33].

### 3.3. The Cardioid Applies also to Capparis Seeds Allowing Description of Intra-Specific Variability

*Capparis spinosa* L. belongs to the family Capparaceae, related to the Brassicaceae in the order Brassicales. Two subspecies are described of *Capparis spinosa* in Tunisia: ssp. *spinosa and* ssp. *rupestris* [39,40]. *C. spinosa* subsp. *rupestris* is more creeping and smaller, and adapted to a variety of diverse conditions including drought and arid conditions in southern regions, where mean annual precipitation is under 100 mm. In addition, seeds are smaller in subsp. *rupestris.* Based on seed shape quantification with the cardiod model, results showed a larger diversity in seed shape in populations of *C. spinosa* subsp. *rupestris* than in ssp. *spinosa* [35].

### 3.4. The Cardioid as a Tool for Seed Shape Quantification in Other Species of Brassicaceae and Legumes

The cardioid or modified cardioid may be useful for seed quantification in many members of the families Brassicaceae and Fabaceae (Figures 9–11). Many species give J index values over 90 with these

models, and in some genera such as *Medicago*, different models can be applied with species resembling the cardioid (Figure 10) and others elongated cardioids (Figure 11) [34].

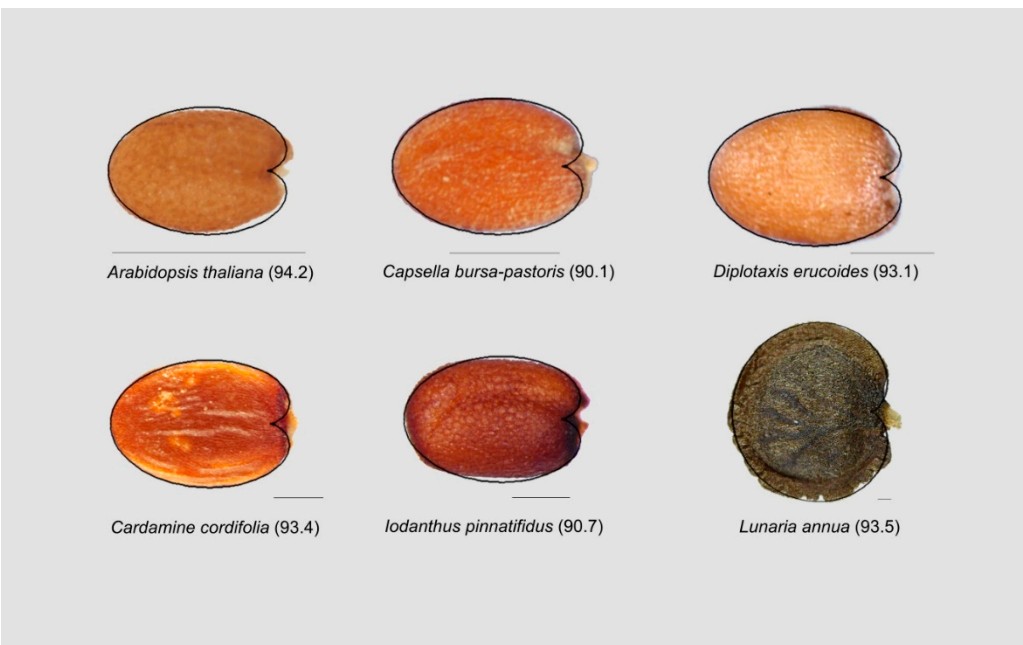

**Figure 9.** The cardioid and modified cardioid (elongated in the horizontal direction) can be used for seed shape quantification in the *Brassicaceae.* The values indicated correspond to J index in the images shown and do not represent mean values for each particular species. Bar equals 0.5 mm.

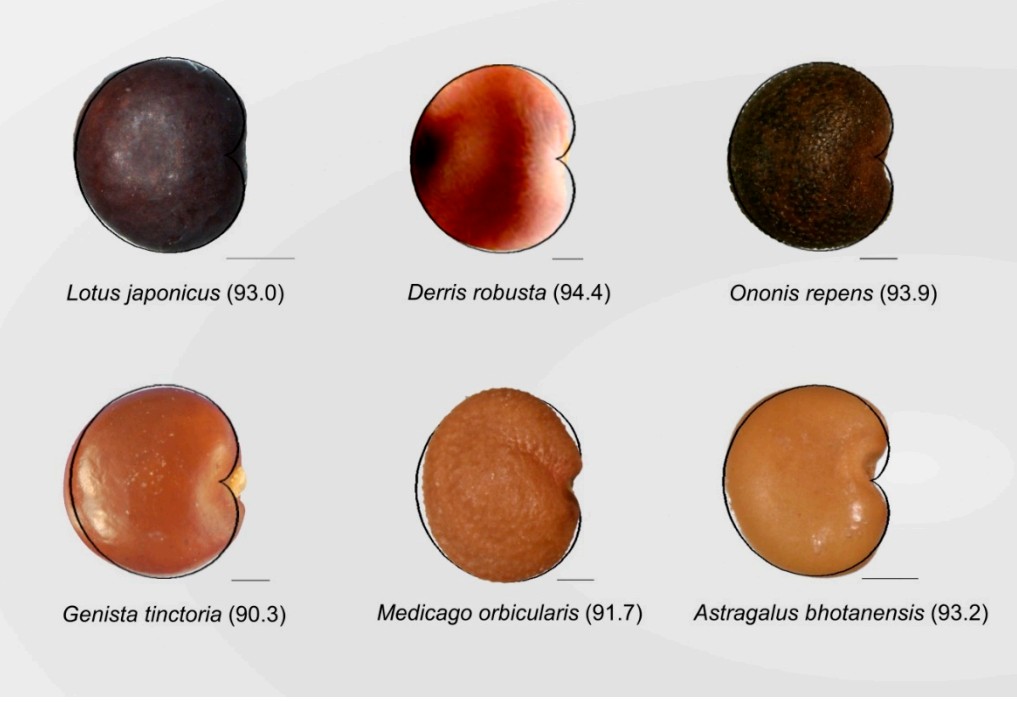

**Figure 10.** The cardioid can be used for seed shape quantification in many species of legumes, including the model species *Lotus japonicus*. The values indicated correspond to J index in the images shown and do not represent mean values for each particular species. Bar equals 0.5 mm.

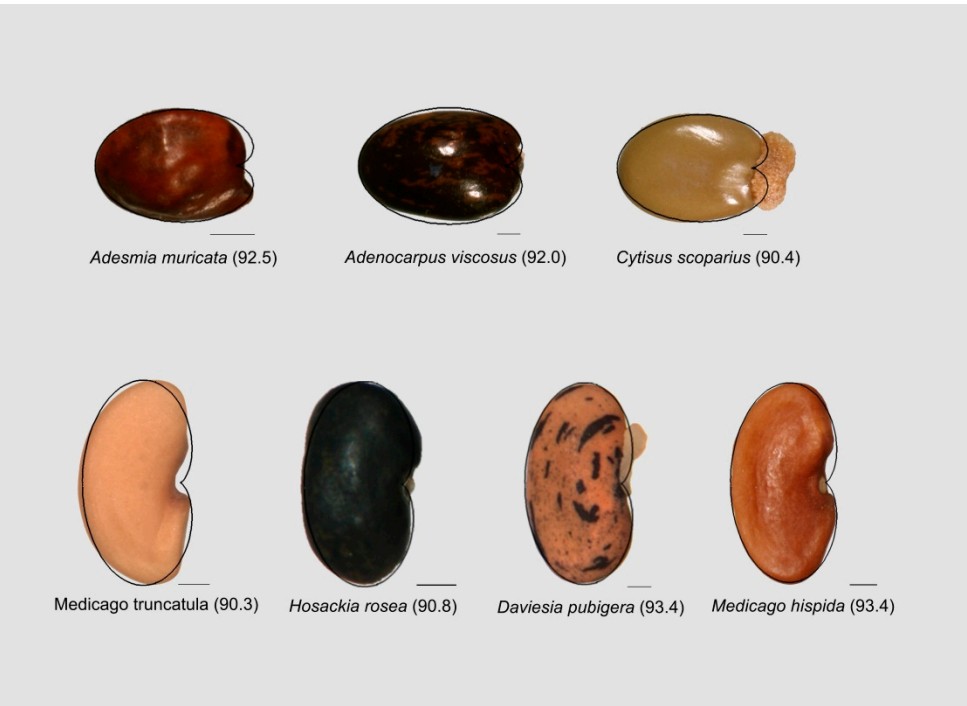

**Figure 11.** The cardioid elongated by a factor of Phi (The Golden Ratio) either in the horizontal axis (top), or in the vertical axis (bottom) is the figure for seed shape quantification in many species of legumes, including the model species *Medicago truncatula*. The values indicated correspond to J index in the images shown and do not represent mean values for each particular species. Bar equals 0.5 mm.

*3.5. The Cardioid as a Model in Seeds of Other Families*

In addition to the observed cases in the Brassicaceae or Capparaceae (Brassicales), and the Fabaceae (Fabales), seeds resembling the cardioid occur in the Aizoaceae and are common in the Papaveraceae (Ranunculales), Malvaceae (Malvales), Caryophyllaceae (Caryophyllales), as well as in other families.

3.5.1. Papaveraceae

The Papaveraceae contain 44 genera with about 770 species, mostly herbs, with only one tree, *Bocconia*, and several shrubs. Seeds of species in the Papaveraceae adjust well to a cardioid (Figure 12) [41].

3.5.2. Malvaceae

In contrast to the Papaveraceae, the family Malvaceae contains herbs, shrubs, and trees. The diverse life forms in the family allow testing our hypothesis of a relationship between life form and similarity to the cardioid in the seeds. In the Malvaceae, seed shape was investigated in a total of 53 genera (118 species) [42]. The greatest resemblance of seeds to a cardioid was found in seeds of herbs (Figure 13, Table 1), whereas less cases of similarity to the cardioid and reduced values of the J index were found in trees. Among the subfamilies of the Malvaceae, the Malvoideae presented most herbaceous species. Out of 77 species analyzed in this subfamily, 53 were herbs and 24 shrubs or trees. The seeds of woody species in this family presented reduced similarity to the cardioid and lower values of the J index than in herbaceous species [42].

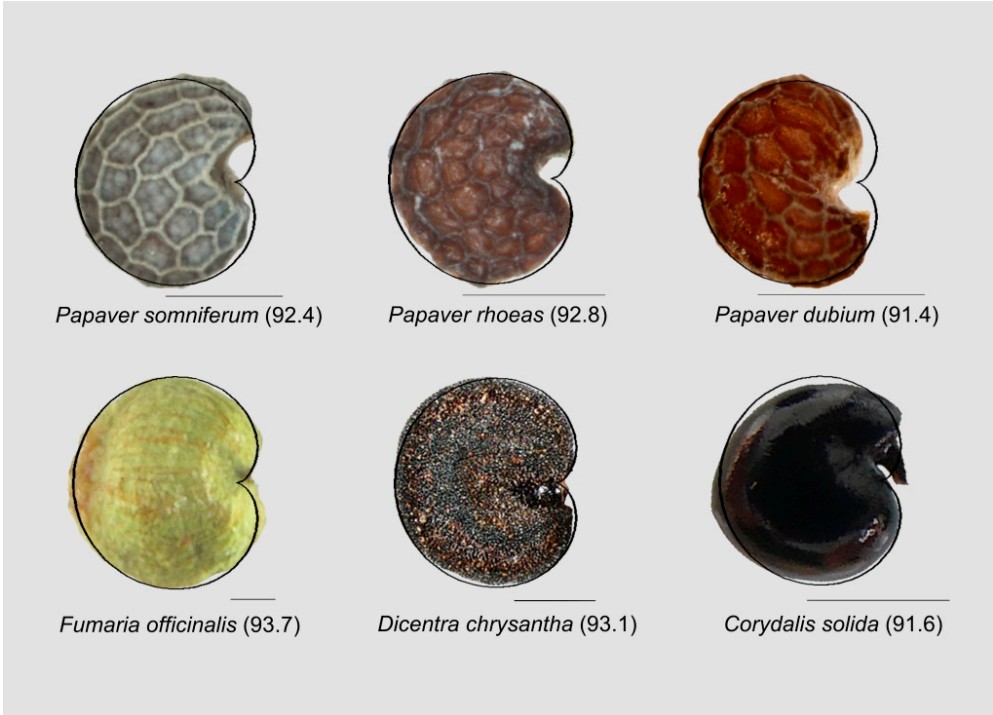

**Figure 12.** The cardioid is a model for seed shape description and quantification in species of the Papaveraceae. The values indicated correspond to J index in the images shown and do not represent mean values for each particular species. Bar equals 0.5 mm.

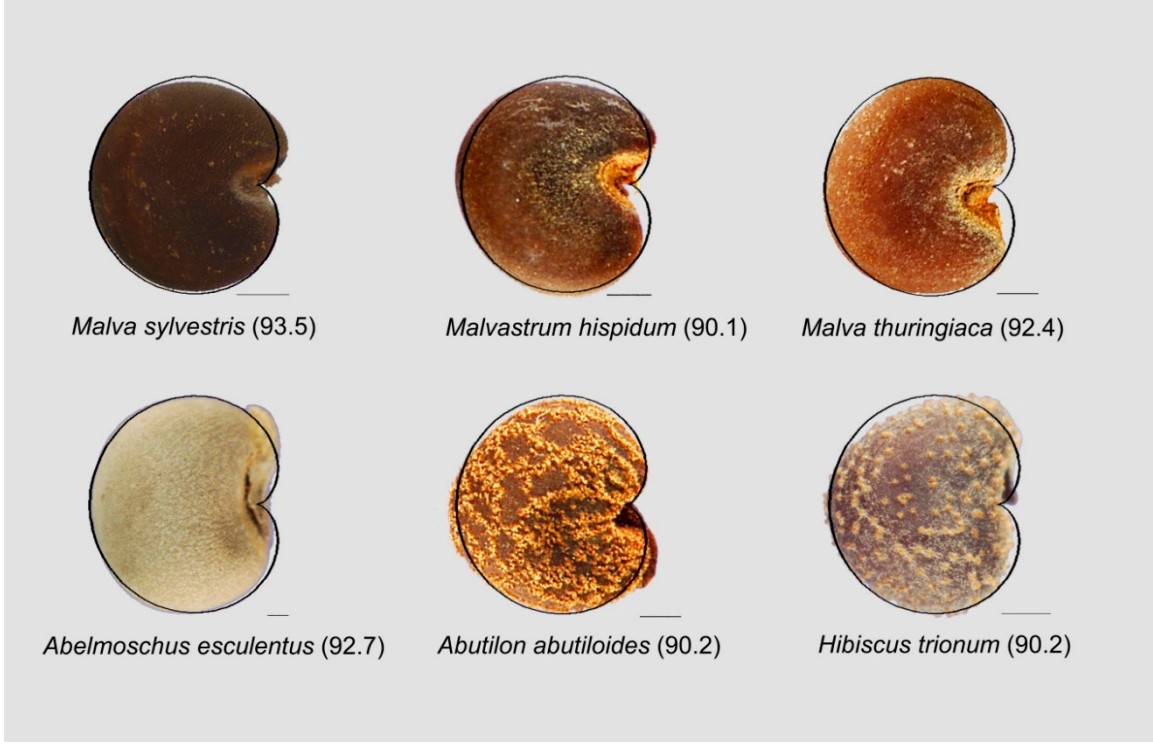

**Figure 13.** The cardioid is a model for seed shape description and quantification in species in the Malvaceae. All the species represented are herbs, except Abutilon, which is a sub-shrub. The values indicated correspond to J index in the images shown and do not mean values for each particular species. Bar equals 0.5 mm.

**Table 1.** Summary of J index values in the Malvaceae. Adapted from [42]. Mean values of the J index were compared by ANOVA (Scheffé's test; IBM SPSS statistics v25) between three groups of species according to their life form (trees, shrubs, and herbs). Different letters indicate significant differences at $p = 0.05$.

| Plant Form | Number of Species | Maximum Value | Minimum Value | Standard Deviation | Mean Values of the Subsets Different for $p = 0.05$ | | |
|---|---|---|---|---|---|---|---|
| Trees | 20 | 87.6 | 50.0 | 10.0 | 70.1 (a) | | |
| Shrubs | 26 | 92.8 | 55.2 | 9.2 | | 79.6 (b) | |
| Herbs | 56 | 92.8 | 66.6 | 5.7 | | | 84.9 (c) |

### 3.5.3. Caryophyllaceae

The cardioid model applies also to species in the Caryophyllaceae, in particular to plant species of small size with short life cycles (Figure 14) [43].

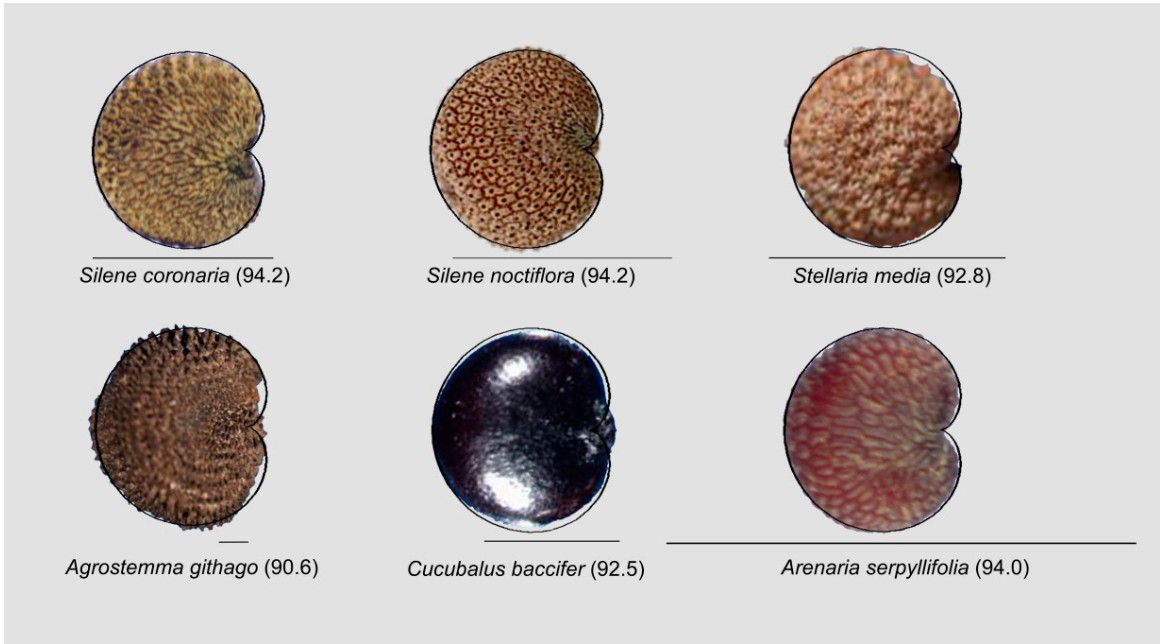

**Figure 14.** The cardioid is a model for seed shape description and quantification in species of Caryophyllales (Caryophyllaceae). The values indicated correspond to J index in the images shown and do not represent mean values for each particular species. Bar equals 0.5 mm.

### 3.6. The Ellipse Is a Model for Seed Morphology in Diverse Families

### 3.6.1. Euphorbiaceae

Morphological seed description of *Jatropha* and *Ricinus* in the Euphorbiaceae allows the identification of variations in the comparison between intra-specific groups.

Seeds of *Jatropha curcas* resemble an ellipse whose proportion between major and minor axes is equal to the Golden Ratio [20]. Comparison of seed shape between nine varieties of *Jatropha curcas* obtained from different origins and grown in Tunisia indicated a relationship between seed yield, size, and shape. Lowest yield was obtained in the accessions with smaller and more circular seeds, whose shape adjusted less well to an ellipse (lower J index values) [28].

Seeds of *Ricinus communis* L. also resemble an ellipse. J index was evaluated in populations of *Ricinus* cultivated in different geographic localizations and found to be lower in the populations grown in the desert [29,30].

### 3.6.2. Oleaceae

Seeds and fruits of species *Olea europaea*, in the Oleaceae, adjust well to an ellipse allowing the comparison between subspecies and varieties [27].

### 3.6.3. Campanulaceae

Seeds in the Campanulaceae and many genera of the Asteraceae (Asterales) resemble the ellipse (Figure 15), but the oval is also frequent in the Asteraceae.

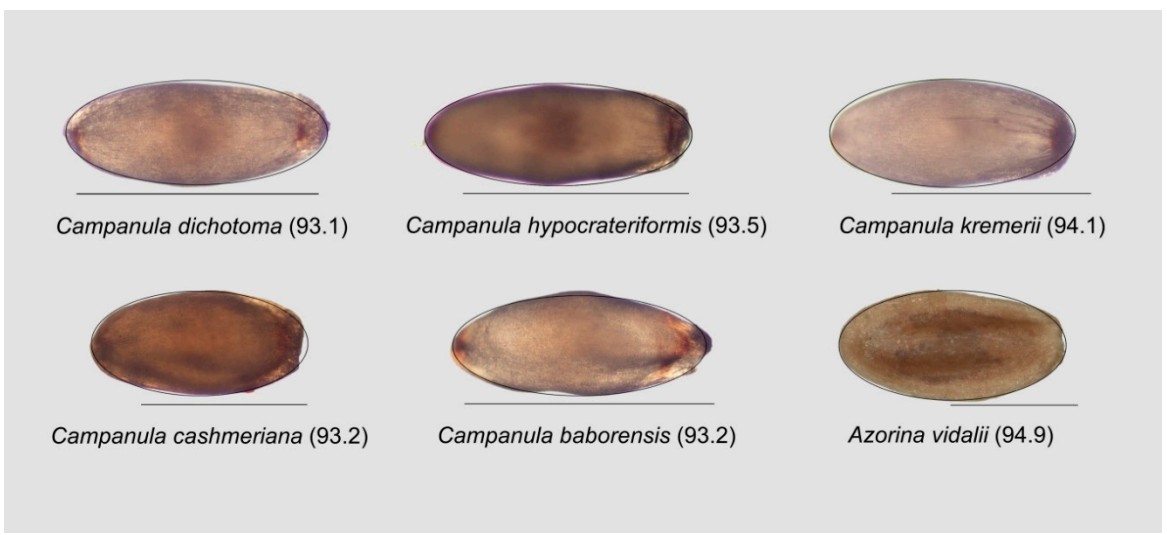

**Figure 15.** Seeds resembling an ellipse (Campanulaceae). The values indicated correspond to J index in the images shown and do not represent mean values for each particular species. Bar equals 0.5 mm.

### 3.6.4. Fagaceae

The ellipse describes well the images of the glands of *Quercus suber*, as well as other species in the Fagales.

### 3.7. The Oval Describes the Bi-Dimensional Image of Seeds in the Cucurbitaceae Well as in Other Families

Seed images of many species in order Cucurbitales adjust well to an oval (Figure 16) [44]. This figure is useful for seed quantification in other families, for example Apiaceae, and may also apply to species in the Amborellaceae (Amborellales), Apocynaceae (Gentianales), Caryophyllaceae (Caryophyllales), Rutaceae (Sapindales), as well as to genera and species in the Asteraceae.

### 3.8. Seed Images Resembling Lenses

The lenses are figures similar to the ellipses but have higher curvature values in their poles [45,46]. Adjust to a lens is good in *Stemonurus* sp. (Stemonuraceae) and *Ilex* sp. (Aquifoliaceae), both in the order Aquifoliales), as well as in some species of the Arecaceae (Arecales) and Calycera (Calyceraceae, Asterales), as well as in the Celastrales.

### 3.9. Other Geometric Figures Useful in Seed Description and Quantification

### 3.9.1. Heart-Shaped Curves

Diverse heart-shaped curves define morphological types in the Vitaceae (Vitales; Figure 17).

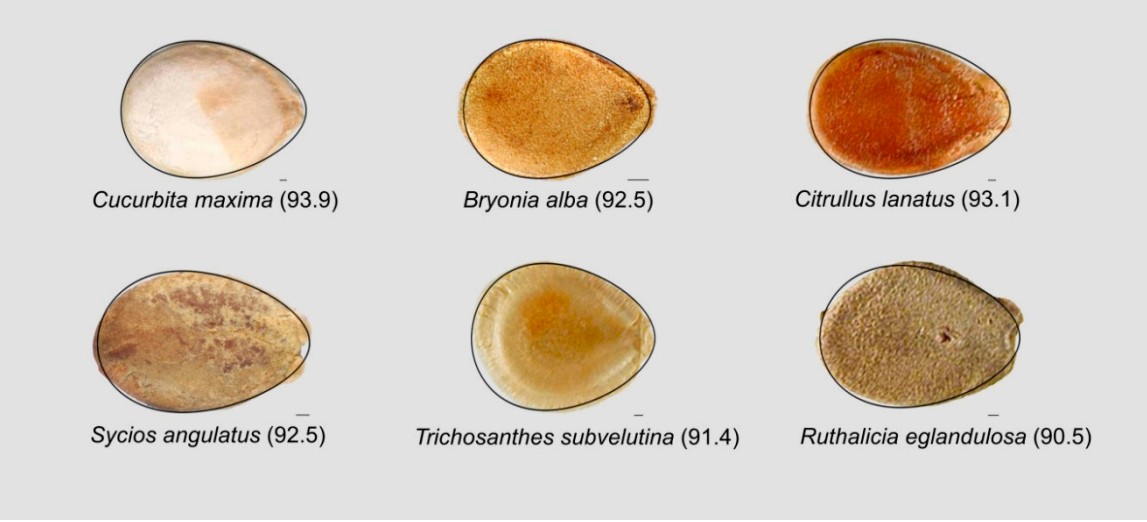

**Figure 16.** Seed images resemble ovals in the Cucurbitaceae. The values indicated correspond to J index in the images shown and do not represent mean values for each particular species. Bar equals 0.5 mm.

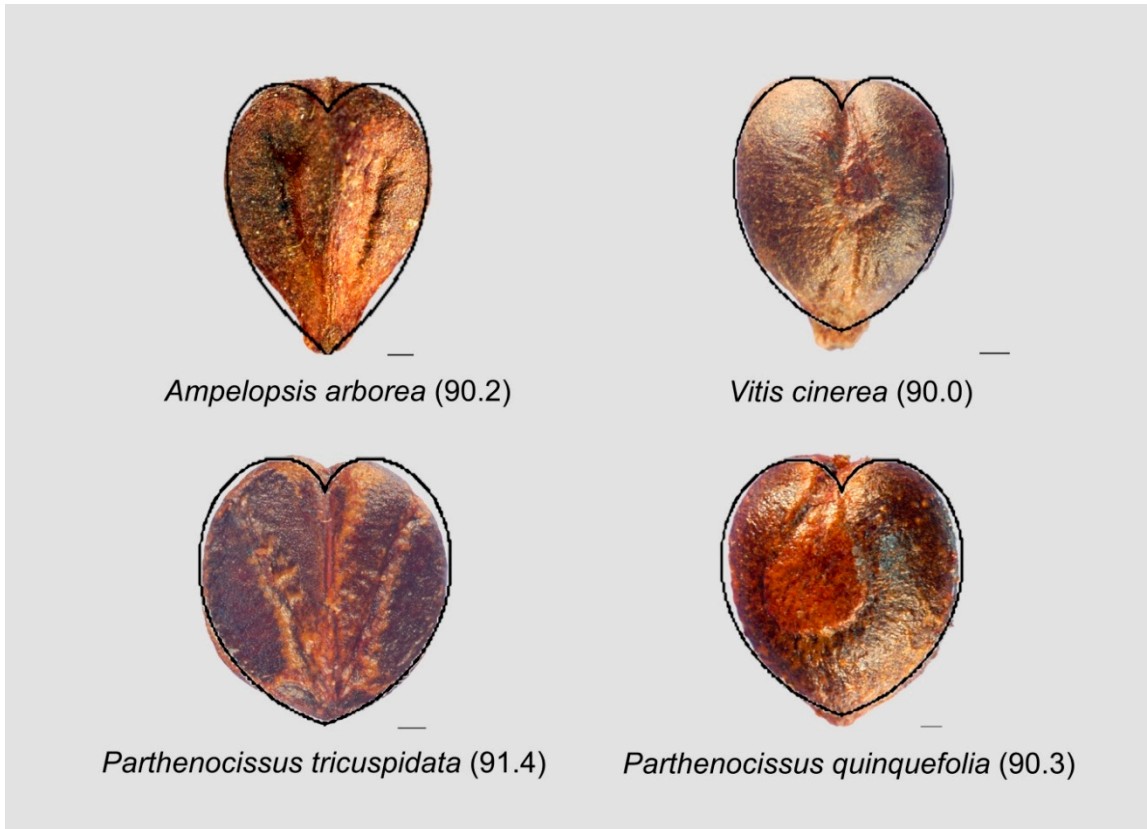

**Figure 17.** Heart-shaped curves in the Vitales. The values indicated correspond to J index in the images shown and do not represent mean values for each particular species. Bar equals 0.5 mm.

3.9.2. Seeds Resembling the Contour of Fibonacci's Spiral

We have found seeds resembling the contour of Fibonacci's spiral in the Coriariaceae (Cucurbitales; Figure 18) as well as in the Alismataceae (Alismatales; Figure 19).

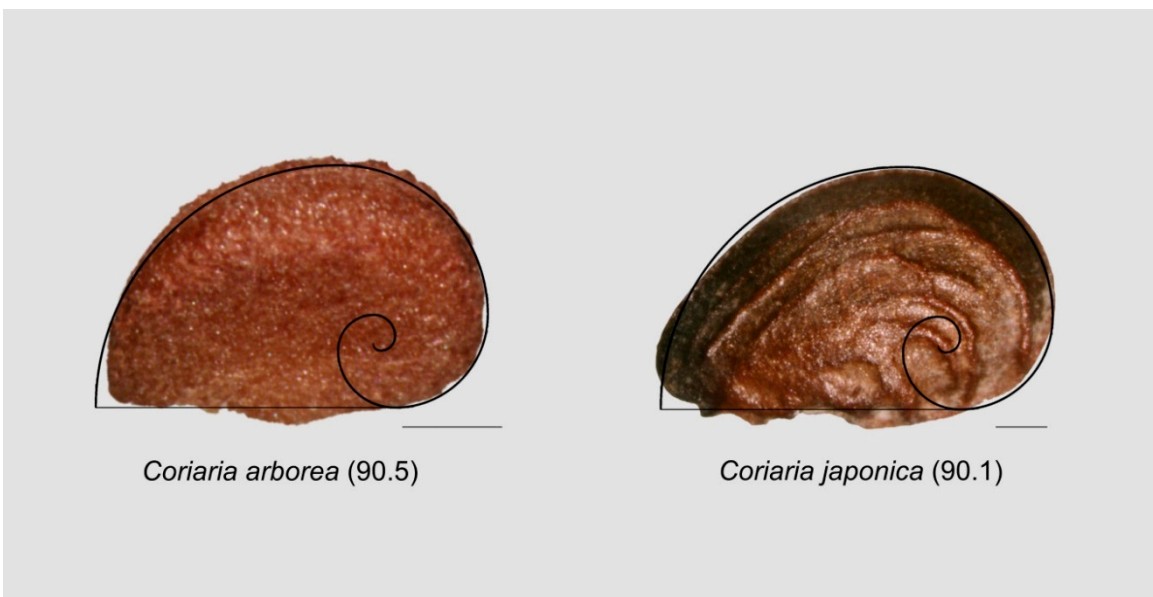

**Figure 18.** Seeds resembling the contour of Fibonacci's spiral in species of *Coriaria*. The values indicated correspond to J index in the images shown and do not represent mean values for each particular species. Bar equals 0.5 mm.

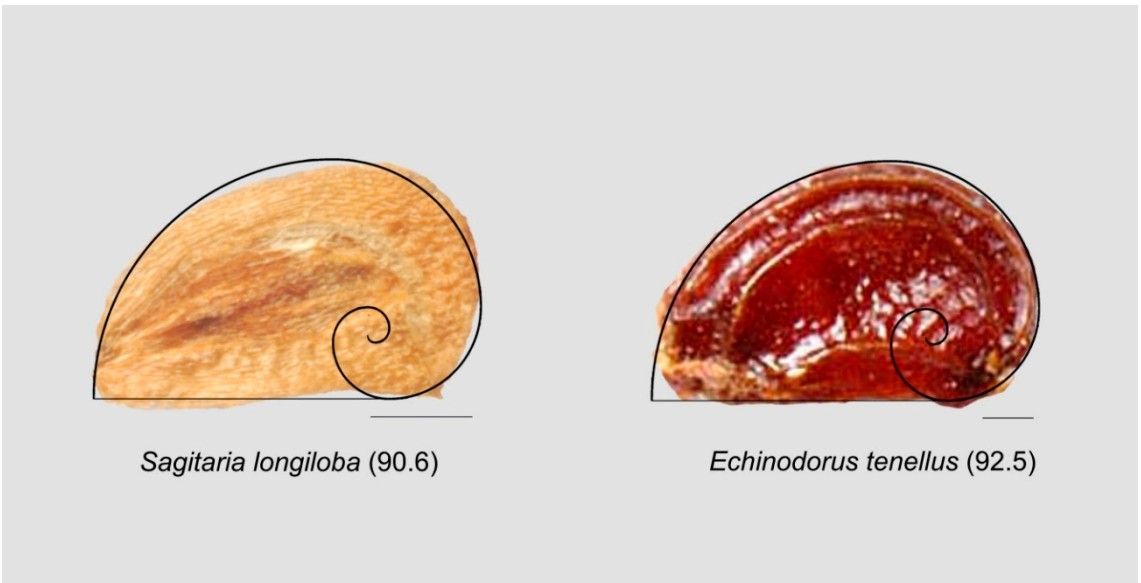

**Figure 19.** Seeds resembling the contour of Fibonacci's spiral in species of the Alismataceae. The values indicated correspond to J index in the images shown and do not represent mean values for each particular species. Bar equals 0.5 mm.

## 4. Discussion

The lack of methods to quantify seed shape results in a difficulty to define shape at the species level, and consequently, to describe with accuracy both intra- as well as inter-specific variations in seed shape.

Variations in seed shape may be due to many reasons, for example: (1) Adaptations for dispersal, (2) structural restrictions and conditions inside the ovary in which the seeds develop, such as in the Nictaginaceae and Nitrariaceae, and (3) other intra-specific variations are due to seed formation in different parts of the inflorescence, or seeds formed under different environmental conditions.

Nevertheless, in general, seed shape is conserved at the species level. The description of shape variations requires a method to quantify seed shape.

Seed shape quantification is more feasible in those cases where seeds resemble geometric figures and whose shape is relatively constant. We have seen many examples of the presence of such geometrically shaped seeds in different families. The seeds of the model species (*Arabidopsis thaliana*, in the Cruciferae; *Lotus japonicus* and *Medicago truncatula*, in the Fabaceae) resemble geometric figures, thus raising the possibility that the conditions of life shared by model species may be related to aspects of the geometry of their seeds. In fact, seeds resembling simple geometric shapes are common in fast-growing annual species in other taxonomic groups such as the Caryophyllales, the Malvales, and the Ranunculales. Among them, cardioid-shaped seeds are common in fast growing species, such as *Silene*, *Spergula*, *Stellaria,* and *Saponaria* in the Caryophyllaceae, as well as many species in the Papaveraceae (Ranunculales).

The genetic control of seed shape is far from being resolved. Species from different taxonomic groups may have a similar seed shape in common, and on the contrary, close relatives in the same family may have different seed shapes. Seeds are forms specialized for survival during long periods of time under difficult conditions such as water and nutrient restrictions. Many metabolic pathways have to be coordinately regulated during seed formation. In their aspect of resistance forms, seeds resemble the pupal phase of insects [47]. In both cases, a complex genetic regulation must be set up involving the coordinated regulation of complex developmental pathways. Work with model systems may be of great help for the identification of the pathways and mechanisms responsible for each type of seed shape. In this context, it is interesting that the *ats* mutant described in *Arabidopsis* has cardioid-shaped seeds [31] instead of the normal shape of an elongated cardioid. However, shape is a complex character and most probably under the control of diverse pathways and it is possible that this phenotype change in *ats* mutants may be obtained by other genetic alterations. For example, mutants in brassinolide pathway also have shortened seeds [48]. Quantitative trait loci analysis is a powerful method to identify genomic components involved in seed shape [49]. The application of this and other modern methods requires the identification of morphological characters independent of size. The work presented here for the description of seed shape opens the way to combine seed morphology with other methods such as visible and near-infrared hyperspectral imaging [50] that may result in the discovery of new relationships in taxonomy and phylogeny.

## 5. Conclusions

(1) A method to define and quantify seed shape is described based on the comparison with geometric models. (2) Geometric models include the cardioid, ellipse, oval, contour of Fibonacci's spiral, lens, and diverse heart-shaped curves. (3) The combination of geometric models with statistical methods allows the quantitative analysis of seed shapes and opens the way to shape description and quantification in seeds of diverse plant families. (4) The method may be useful to discover new relationships in taxonomy and phylogeny as well as in the description of mutants and shape modifications in diverse growth conditions.

**Author Contributions:** Conceptualization, E.C.; methodology, J.J.M.G.; software, J.J.M.G.; formal analysis, J.J.M.G.; investigation, E.C. and J.J.M.G.; data curation, E.C. and J.J.M.G.; writing—original draft preparation, E.C.; writing—review and editing, E.C.

**Funding:** This research received no external funding

**Acknowledgments:** Seeds of the Campanulaceae in Figure 15 were kindly provided by Juan José Aldasoro and Marisa Alarcón.

**Conflicts of Interest:** The authors declare no conflict of interest.

## Appendix A. Origin of the Images Used in the Figures

The images whose precedence is not mentioned belong to our image collection at IRNASA-CSIC.

Figure 2:

Cardioid: like2do.com

Cyclamen: https://wildflowerfinder.org.uk/Flowers/S/Sowbread(Eastern)/Sowbread(Eastern).htm. Photo: © Angela Stephens.

Embryo: http://www.meihsieh.com/meid2go/hereissomething-globalsoftpirka, https://creepypasta.fandom.com/wiki/Creepypasta_Wiki:Creepy_Images/Page_52

Figure 3:

Paramecium: https://www.asturnatura.com/especie/stylostomum-ellipse.html

Figure 4:

Argonaute: https://www.3djuegos.com/comunidad-foros/tema/6199248/0/argonauta-argonauta-argo/

Figure 7: Sole from: https://conxemar.com/es/lenguado-europeo

Figure 9: Diplotaxis erucoides, Cardamine cordifolia, Lunaria annua from USDA. Iodanthus pinnatifidus from University of Minnesota (TMI).

Figure 10: Genista tinctoria, Derris robusta from USDA. Ononis repens, Astragalus bhotanensis from herbarium (PE).

Figure 11: Adesmia muricata, Adenocarpus viscosum, Medicago hispida from herbarium (PE). Daviesia pubigera from RBG Kew.

Figure 12: *Dicentra chrysantha* from USDA.

Figure 13: Malvastrum hispidum, Lavatera thuringiaca, Abelmoschus esculentus, Abutilon lignosum, Hibiscus trionum, from USDA.

Figure 14: Silene coronaria, Silene noctifolia from USDA. Stellaria media, Agrostemma ghithago, Cucubalus baccifer, Arenaria serpyllifolia from herbarium (PE).

Figure 16: *Cucurbita maxima* from the Department of Horticulture and Crop Science (The Ohio State University); *Bryonia alba*, *Citrullus lanatus*, *Scyos angulatus*, *Trichosanthes subvelutina*, *Coccinia eglandulosa* from USDA.

Figure 17: Ampelopsis arborea, Vitis cinerea, Parthenocissus tricuspidata, Parthenocissus quinquefolia from USDA.

Figure 18: *Sagitaria longiloba* (Alismataceae) and *Echinodorus tenellus* (Alismataceae) from USDA; *Coriaria arborea* and *Coriaria japonica* from herbarium (PE).

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
