# Peer review of "Seed Shape Description and Quantification by Comparison with Geometric Models"

_horticulturae, doi:10.3390/horticulturae5030060_

Round 1
Reviewer 1 Report
Seed shape description and quantification by comparison with geometric models.
The manuscript describe the geometric models which may be applied for seed shape quantification in diverse plant families and discuss the relationship between seed shape and other plant characteristics and the potential of seed morphology in taxonomy.
1. The structure of the manuscript corresponds with the standards that apply to original research papers, and no corrections are needed.
2 The title is adequate to the content of the paper and are correctly formulated.
3. The abstract is sufficiently informative, properly reflects to the content of work.
4. Introduction includes a proper introduction to the presented results. The objectives of the study are clearly presented and supported with the most recent knowledge on the subject. It is the evidence of the very good orientation in the problem discussed in the paper.The research goals were given clearly.
5. Manuscript are easy to follow as the chapter is divided in some parts. All of them are properly presented, comprehensively discussed and tightly argued as well as supported with adequate source material in the form of tables and figures.
6. Discussion is multithreaded, closely refers to the presented results, and confronts them with findings made by other researchers. It is the evidence of the very good orientation in the problem discussed in the paper. The most important works are cited.
7. The cited literature is relevant and creatively incorporated into the text. In the manuscript the most important works are cited.
8. All tables and figures are clear, not overloaded with data well-explained in the captions and give a good documentation of the results.
9. The manuscript is well written in clear and concise manner, and the style is in general correct.
The article is highly interesting, because of the chosen subject. In my opinion the paper can be accepted for publication in Horticulturae.
Specific comments
Key words should be partially changed and supplemented.
Please revise your manuscript carefully and sure that all bibliographic items are cited.
Conclusions – should be formulated.
Author Response
Dear Reviewer,
Thank you very much for your comments that contribute to improve the quality of the article. They have been taken into account for the new version:
Key words have been partially changed and supplemented.
The manuscript has been carefully revised confirming that all bibliographic items are cited.
Conclusions have been formulated.
Reviewer 2 Report
This paper is interesting and describe the geometric models that may be applied for seed shape quantification in diverse plant families.
In my opinion it requires a major revision before be suitable for publication, addressing the following aspects:
The manuscript should be revised by a speaker native
The work is interesting and it represents a step forward in the scientific literature.
The topic is suitable for the publication on the Horticulturae Journal.
Considerations about the sections:
1. The introduction does not report the principal pieces of information about the topic. I think that a subsection about works performed from visible and near-infrared (VIS/NIR) hyperspectral images for the classification of seed could be included
The manuscript do not objectively reflect the most significant findings in a Conclusion section. Also, the authors should explain the challenges and opportunities about the future research in this topic.
Author Response
Dear Reviewer,
Thank you very much for your comments that have contributed to improve the quality of the article. They have been taken into account for the new version:
English language has been reviewed.
The introduction has been corrected.
A commentary about visible and near-infrared (VIS/NIR) hyperspectral images has been included in the discussion. A reference has been added related to this topic:
He, X.; Feng, X.; Sun, D.; Liu, F.; Bao, Y.; He, Y. Rapid and Nondestructive Measurement of Rice Seed Vitality of Different Years Using Near-Infrared Hyperspectral Imaging. Molecules 2019, 24(12), 2227; https://doi.org/10.3390/molecules24122227
A conclusions section has been added.
Also, challenges and opportunities about the future research in this topic are now commented.
Reviewer 3 Report
The document exposes a novel seed research area to develop and holds an interesting perspective that glimpses at many potential practical uses. However, the document, in general, is not comprehensive enough for a review of seed diversity and has not a pattern or organization. The first of these issues does not seem problematic due to the novelty and apparent lack of references in the area besides those from the same authors. The second is more critical and should be attended before publication. Otherwise, the document in general looks like a collection of disperse examples and guestimates with great photographs.
The criteria used to narrow down this review could be explained. Only 25 references seem relevant, out of which 15 are self-citations that do not appear as very extensive for a review unless the criteria are restricted, and restrictions clarified. I wonder if there are enough important references available to make a worthy review.
Lines 14-15: We describe six geometric figures
Lines 15-16: ...in the family of horticultural plants…. The document neither includes all the families of the mean horticultural plants globally neither is restricted to only horticultural plants. Could you use a fitting subgroup or category that either comprises all the families in the paper or numbers the horticultural families used?
Line 30: … sphere, oval …and others. Spheres are not included and there are only six shapes explored not including the sphere (or circle) that is actually a common seed shape. Some light could be given on the reasons for selecting these particular shapes over all the possible ones or if these are the only ones available.
Line 45: as in lines 16-16
Line 48: delete ‘that may be’
Lines 49-55: as line 30
Lines 57-86: Shape descriptions are long, redundant and unnecessary, considering they are perfectly illustrated in the figures and described again in the figures’ captions
Figures 2, 3, 4, 5, 6 and 7: three images per figure are excessive and irrelevant for a scientific review document in a specialised area with a clear written description
Captions of figures 2, 3, 4 and 7 the content of the figure is described twice, once in the text and then in the caption. Please, review this
Lines 51, 54, 92, 94: Fibonacci’s spiral, golden spiral are fine, but consistency on a single term along the text would be better. Also, stress is missing on the fact that is only the contour and not the spiral per se the reference shape for seeds.
Lines 124-130, none of the examples pictured is a horticultural species nor has shapes reviewed on this document. I fail to see the use of the image and extended explanation on these examples.
Lines 145-157: Reads almost like a methodology but requires a clearer description of the process
Line 153: Delete ‘will’
Line 205: How many are ‘many’?
Lines 214-215 The 118 species of the Malvaceae investigated were investigated by ..? is all the paragraph on this page referring to reference 42?
Lines 215-215 “The greatest resemblance of seed…. Trees” it should be marked in figure 13 what is a tree or herb is or with numbers, what range of J values are associated with a herb shrub or tree. A graph or table could be useful to present those values and trends. It would be also useful to include such numbers in the other shapes or families.
Lines 224-225 “... do not represent mean values for each particular species” What does it represent instead?
Lines 237-238 Seeds of Jartropa curcas.. Golden Radio. Is Golden Ration defined somewhere in the document? Please explain if relevant or delete.
Author Response
Thank you very much for your commentaries. These have been taken into consideration in the preparation of a new version.
The following indications concern the main commentaries, the remaining questions have been corrected in the text:
Q1. The document neither includes all the families of the mean horticultural plants globally neither is restricted to only horticultural plants. Could you use a fitting subgroup or category that either comprises all the families in the paper or numbers the horticultural families used?
A1. We have indicated now:
First we describe six geometric figures that may be used as models for shape description and quantification and later on, we give an overview with examples of some of the types of seed morphology in angiosperms including families of horticultural plants and addressing the question of how is the distribution of seed shape in these families.
Q2. Concerning these two commentaries:
Lines 57-86: Shape descriptions are long, redundant and unnecessary, considering they are perfectly illustrated in the figures and described again in the figures’ captions
Captions of figures 2, 3, 4 and 7 the content of the figure is described twice, once in the text and then in the caption. Please, review this
A2. Figure legends need to be self-explanatory. For this reason it is often observed in the scientific literature that information is redundant in figure legends and text.
Q3. Concerning this:
Figures 2, 3, 4, 5, 6 and 7: three images per figure are excessive and irrelevant for a scientific review document in a specialised area with a clear written description.
A3. We consider important to leave the images with three parts as they are.
Q4. Lines 51, 54, 92, 94: Fibonacci’s spiral, golden spiral are fine, but consistency on a single term along the text would be better. Also, stress is missing on the fact that is only the contour and not the spiral per se the reference shape for seeds.
A4. The contour of Fibonacci’s spiral has been indicated through the text.
Q5. Lines 124-130, none of the examples pictured is a horticultural species nor has shapes reviewed on this document. I fail to see the use of the image and extended explanation on these examples.
A5. The first condition I asked to Sharon Wang, the editor of this special issue, before accepting my participation on it was that the term Horticulture should be used in a broad sense. This includes arboriculture and medicinal herbs as a part of horticulture. The images represent an important medicinal plant (Peganum harmala) as well as two important trees for arboriculture.
Q6. Lines 145-157: Reads almost like a methodology but requires a clearer description of the process.
A6. A clear description of the process can be found in the references given.
Q7. In the paragraph concerning the Malvaceae:
Lines 214-215 The 118 species of the Malvaceae investigated were investigated by ..? is all the paragraph on this page referring to reference 42?
A7. Yes. All the paragraph is referring to reference 42, and it has been now indicated.
Q8. Lines 215-215 “The greatest resemblance of seed…. Trees” it should be marked in figure 13 what is a tree or herb is or with numbers, what range of J values are associated with a herb shrub or tree. A graph or table could be useful to present those values and trends. It would be also useful to include such numbers in the other shapes or families.
A8. Figure 13 contains seeds of herbs and this has been now indicated in the text. A table has been added with this information (Table 1).
Q9. Lines 224-225 “... do not represent mean values for each particular species” What does it represent instead?
A9. As indicated: The values indicated correspond to J index in the images shown.
Q10. Lines 237-238 Seeds of Jartropa curcas. Golden Radio. Is Golden Ration defined somewhere in the document? Please explain if relevant or delete.
A10. A reference is now given.
Round 2
Reviewer 2 Report
Accept in the present form
Author Response
Thank you for your comments.
Reviewer 3 Report
Thanks for considering the previous comments for changes.
The document has been significantly improved. Although I would have liked to see a reason as to why leaving the three repetitive images that would look more appropriate elsewhere than a scientific journal. Also a few more pertinent references, not self-citations.
Low evaluation in question three is only due to the nature of the publication (a review)
Looking forward to reading new research of yours in the area including more shapes, species and relationships in taxonomy and phylogeny.
Author Response
Dear Reviewer,
Thank you for your commentaries.
We prefer to leave three images por figure including an image showing how the geometric figures used as models can also be applied to animal systems. In figure 1 we present a summary of the models used in the description and quantification of seeds, but it is important to remark also visually that these geometric figures can also be applied as models for the description and quantification of animal shapes.
We made a review of the method such as it has been set up and applied and in so far, this corresponds mainly to work published by our laboratory. Nevertheless, other references concerning general aspects as well as related publications have been taken into account when required.
With best regards.